# Pentacyclic Triterpenoid Phytochemicals with Anticancer Activity: Updated Studies on Mechanisms and Targeted Delivery

**DOI:** 10.3390/ijms241612923

**Published:** 2023-08-18

**Authors:** Madalina Nistor, Dumitrita Rugina, Zorita Diaconeasa, Carmen Socaciu, Mihai Adrian Socaciu

**Affiliations:** 1Department of Biochemistry, University of Agricultural Sciences and Veterinary Medicine, 400372 Cluj-Napoca, Romania; nistor.madalina@usamvcluj.ro (M.N.); dumitrita.rugina@usamvcluj.ro (D.R.); zorita.sconta@usamvcluj.ro (Z.D.); 2Department of Biotechnology, BIODIATECH—Research Centre for Applied Biotechnology in Diagnosis and Molecular Therapy, 400478 Cluj-Napoca, Romania; 3Department of Radiology, Imaging & Nuclear Medicine, Faculty of Medicine, University of Medicine and Pharmacy “Iuliu Hatieganu”, 400347 Cluj-Napoca, Romania

**Keywords:** pentacyclic triterpenoids, betulinic acid, anticancer activity, liposomes, nanolipid complexes, microbubble conjugates, in vivo sonoporation

## Abstract

Pentacyclic triterpenoids (TTs) represent a unique family of phytochemicals with interesting properties and pharmacological effects, with some representatives, such as betulinic acid (BA) and betulin (B), being mainly investigated as potential anticancer molecules. Considering the recent scientific and preclinical investigations, a review of their anticancer mechanisms, structure-related activity, and efficiency improved by their insertion in nanolipid vehicles for targeted delivery is presented. A systematic literature study about their effects on tumor cells in vitro and in vivo, as free molecules or encapsulated in liposomes or nanolipids, is discussed. A special approach is given to liposome-TTs and nanolipid-TTs complexes to be linked to microbubbles, known as contrast agents in ultrasonography. The production of such supramolecular conjugates to deliver the drugs to target cells via sonoporation represents a new scientific and applicative direction to improve TT efficiency, considering that they have limited availability as lipophilic molecules. Relevant and recent examples of in vitro and in vivo studies, as well as the challenges for the next steps towards the application of these complex delivery systems to tumor cells, are discussed, as are the challenges for the next steps towards the application of targeted delivery to tumor cells, opening new directions for innovative nanotechnological solutions.

## 1. Introduction

Pentacyclic triterpenoids [TTs] represent a unique family of phytochemicals or plant secondary metabolites that possess a 30-carbon skeleton with five interconnected rings. This molecular family includes terpenes with hydrocarbon cycles or hydroxylated carboxylate derivatives and, depending on their molecular structure, are divided into five classes, the most known being lupane-type (lupeol, betulin, and betulinic acid), oleanane-type (oleanolic acid, ß-amyrin, and erythrodiol), and ursane-type (ursolic acid, α-amyrin, and uvaol).

The most significant lupane-type TT molecules and their structures are illustrated in Figure 1.

Pentacyclic TTs are naturally found in a great variety of fruits, vegetables, and medicinal plants, acting as defensive compounds against pathogens. These are the subjects of numerous phytochemical and pharmacological studies. The interest in these molecules has increased since 1995, when B and BA cytotoxicity against human melanoma lines in vitro and in vivo was shown [1,2]. The advances in their chemical modifications and effective delivery vehicles to improve bioavailability and bioactivity, as well as the molecular mechanisms to explain their activity, were also described [3]. As the best representatives of this family, numerous reviews have been dedicated to their effects in more than 3200 publications related to their isolation from natural sources and cytotoxic effects, including patents and patent applications [4,5,6,7,8,9]. Edible pentacyclic triterpenes (TTs) have proven to have various biological functions, including antiviral, anticancer, antioxidant, anti-inflammatory, and hypoglycemic activities, assuming their action as pharmacological agents and potent anticancer molecules [5,10,11,12,13,14,15]. However, their properties of strong hydrophobicity, poor permeability, poor absorption, and rapid metabolism result in low oral bioavailability, which dramatically hinders their efficacy for use. A recent review systematically summarized the chemical structures, plant sources, bioactivities, absorption, metabolism, and oral bioavailability of TTs and emphasized their self-assembly properties and emerging role as functional delivery carriers for nutrients, suggesting that their derived nanostructures are not only efficient as oral forms introduced into foods but also functional delivery materials for nutrients [16].

## 2. Isolation, Physical, and Chemical Characteristics

Due to the large natural occurrence of B as a BA precursor, many phytochemical studies were focused on plant families with higher levels of B, such as *Betulaceae*, *Platanaceae*, *Dilleniaceae*, *Rhamnaceae*, *Rosaceae*, and *Fagaceae*, including *Ziziphus* spp. *(Rhamnaceae), Syzygium* spp. *(Myrtaceae), Diospyros* spp. *(Ebenaceae)*, and *Paeonia* spp. (*Paeoniaceae*) [5]. The isolation, synthesis, and derivatization of B, BA, and 23-hydroxy-BA, as well as the current knowledge on the pharmacological mechanisms of these compounds and their ability to enact potential antitumor activity, were recently reviewed [8].

One of the most widely reported sources of B and BA is the birch tree bark (*Betula* spp., *Betulaceae*), from which these molecules are extracted in substantial quantities by different procedures on a small or larger scale. Updated and detailed information on botany, traditional uses, phytochemistry, and pharmacological and toxicological research associated with *Betula* species is provided [17]. Recently, the phytochemical profiling of the stem bark of *Betula utilis* from different geographical regions (forestry areas of the Western Carpathians or India) was reported using advanced analytics, mainly UHPLC–ESI-MS/MS [18,19].

Recently, different methods useful for developing a technology for isolating and purifying TTs from birch bark in large quantities were reviewed [7], including details about the B isolation. Effective methods of B isolation and purification from birch bark were recently reviewed, underlying the importance of its purity for scientific and medicinal use [20]. Since B is available from many plant species, especially from birch-kraft pulp, semi-synthetic methods to produce BA from B were also developed to improve the pharmacological effects of BA and optimize its activity [4,6]. BA is found beside B in natural sources but can also be synthesized by B oxidation, amination [amide and amine derivatives], esterification, sulfonation, and alkylation. A multitude of extraction and isolation schemes using chloroform and methanol to separate either the aglycon or the glycosylated derivatives have been used for BA, with or without preliminary defatting with hexane and other related triterpenoids [4]. BA is a white crystalline solid highly soluble in pyridine and acetic acid, with limited solubility in organic alcohols such as methanol, ethanol, chloroform, and ether and low solubility in H_2_O, petroleum ether, DMSO, and benzene. Solubilization of BA in a microemulsion system including methyl acetate/Tween 80 was also reported [21].

## 3. Biological and Pharmacological Activity

The structural features of TTs are related to a wide range of biological actions and a great potential for numerous therapeutic benefits [11]. Especially BA and its derivatives extracted from birch extracts have attracted more attention in the last decades because of their broad spectrum of biological activities, especially their anticancer, antimicrobial, antirheumatic, anti-inflammatory, and wound-healing properties [22]. Mainly the biological activities and pharmacological effects of BA and B, but also L, were reviewed in the last few years, showing diverse pharmacological effects [7,9,23] and their capacity to revitalize the cellular antioxidant, anti-inflammatory, and anti-apoptotic mechanisms. A preliminary pharmacokinetic study of B, extracted from the outer bark of birch (*Betulae alba cortex*), has shown that it is bioavailable and forms an oleogel that is successful in treating actinic keratosis [24]. A pharmacokinetic study of a triterpene-rich dry extract extracted from the outer bark of birch bark extract containing B, BA, OA, L, and erythrodiol was subjected to subchronic toxicity studies on rats using daily ip. doses of up to 540 mg/kg for 28 days. No toxic symptoms or mortality were reported, and no histopathological changes were also reported [24], as was previously found for BA at 500 mg/kg body weight in mice [1].

Regarding BA absorption, distribution, and pharmacokinetics, especially in pre-clinical models, it was shown that, after intraperitoneal administration, BA reaches peak serum levels in a brief span and disperses broadly in the body, reaching its highest concentration in perirenal fat and lower concentrations in the ovaries, spleen, and mammary gland. The elimination half-life ranges around 11–12 h, demonstrating its sustained presence in the body [25].

To elucidate the metabolic fate of BA in humans, microorganisms were utilized as in vitro model systems to predict its potential metabolites based on the extensive homology between microbial and mammalian metabolic pathways. When incubated with resting-cell suspensions of different microorganisms (*Cunninghamella* spp., *Cunninghamella elegans, Bacillus megaterium, and Mucor mucedo*), a series of BA-oxidized and conjugated metabolites were identified [4].

The involvement of different TTs in cellular metabolism and their activity on different metabolic pathways, with consequences for their pharmacological activity, are synthesized in Table 1.

### 3.1. Antiviral, Immunotropic, and Antimicrobial Activity

The potential therapeutic applications of BA for human immunodeficiency virus-1 (HIV-1) infection have led to an increased focus on its pharmacokinetic behavior in mammals. In search of herbal medicinal molecules capable of fighting HIV infection, since 1994, the antiviral properties of *Syzygium claviflorum* leaf extract, including BA and its derivative platanic acid, have been studied. Further research on a semi-synthetic derivative of BA, namely [3-O-[3′,3′-dimethylsuccinyl] BA, known under the commercial name Bevirimat, showed its stronger antiviral properties [41]. Also, the synthesis and structure–activity relationships of triterpene derivatives as anti-HIV agents were reported recently [42]. BA and B derivatives interfere with the viral life cycle by targeting a variety of mechanisms associated with viral entry and maturation [43]. Their inhibitory effects on HIV-1 reverse transcriptase at micro- and nanomolar levels comparable to those of clinically used drugs were also reported [44].

Another review focused on natural and modified BA, UA, and echinocystic acid derivatives as potential anti-HIV and antitumor agents [45]. Many reports indicate that TTs are capable of inhibiting HIV-1 replication. Using an in vitro model system, BA exhibited an IC_50_ value of 1.4 mM, and subsequent derivatives improved its anti-HIV activity [4]. New derivatives of B and BA can be considered precursors of new drugs, which have been tested in phase I, II, or III of clinical trials as well as in clinical experiments on animals with natural cancers, demonstrating their effects as antiviral agents [46]. Lupane-type TTs are also selective virus entry inhibitors in hepatitis B and D [47].

Lupane-type pentacyclic TTs have been demonstrated to have beneficial effects on the immune system [48,49]. Also, many TTs, including B and BA, have been investigated for anti-inflammatory properties in in vitro and in vivo model systems, showing a moderate effect attributed to the inhibition of non-neurogenic pathways and interaction with glucocorticoid receptors [4,50].

Molecules like BA, UA, and especially OA, free or derivatized, showed prominent antibacterial activity against *Staphylococcus aureus* and *Bacillus subtilis,* while *Escherichia coli* showed resistance [51,52]. More recently, six TTs (L, B, BA, OA, UA, and 3-hydroxy-11-ursen-28,13-olide) were evaluated and isolated from the *Alstonia scholaris* leaf extract. After identification by NMR, their antibacterial activity was demonstrated to be synergistic with antibiotics against bacterial pathogens. Both OA and UA showed antibacterial activity limited to Gram-positive bacteria, while UA showed a synergistic effect with ampicillin and tetracycline against both *Bacillus cereus* and *S. aureus* [53].

BA was found to possess antinociceptive, antihelmintic, and anti-HSV-1 activities [3], while semisynthetic BA and B derivatives have antileishmanial activities [54]. New investigations with Azepane, which has a Lupane Type A-Ring, showed its antimycobacterial activity and were being recommended as a drug candidate [55]. Antibacterial TTs from *Dillenia papuana* and their structure–activity relationships were also reported [56].

### 3.2. Anticancer Activity

The most studied activity of TTs is related to their anticancer action and the perspective of using them in cancer prevention and treatment. Recently, the current status and future prospects related to these aspects were reviewed [57]. Their role and mechanisms involved in the cytotoxic, antiproliferative, and chemosensitizing activity were presented in comprehensive reviews, focusing especially on B and BA, as well as derivatives, as potent anticancer agents and their role in chemoprevention [4,10,12,43,58,59,60,61].

Recent updates on the anticancer efficacy of bioactive pentacyclic TTs via signaling pathways for selective apoptosis were reviewed [62]. The induction of apoptosis was reported through the regulation of BCL-2 and BH3 family proteins, modulation of the inflammatory pathway, interference with cell invagination, and inhibition of metastasis. Nanotechnology is revealed to have an important impact by modifying their physical form for increased solubility and bioavailability, and it is one of the possible ways to mitigate this issue with the help of and modification of their physical forms. BA became the most interesting chemotherapeutic agent, especially after its inclusion in the National Cancer Institute’s RAID program, due to its antitumor properties and ability to induce apoptosis, especially against melanoma [25]. Table 2 includes recent data related to the anticancer effects of BA and B.

Structure–cytotoxicity relationships of these compounds and the impact of purity on the antitumor potency of BA derivatives are also important issues, indicating a positive relationship between B purity and the decrease in cell survival [20]. There is a growing importance of triterpenoids (especially BA and B) as a source of medication for various chronic diseases, and several pieces of evidence suggest that both natural and synthetic triterpenoids have potential anticancer activities, as reviewed in experimental animal models [57]. Emulsions containing 80% B and 10% BA were tested in phase I, II, or III of clinical trials and experiments on animals (mice and rats) and showed toxic symptoms at doses above 500 mg/kg [46].

Currently, the mechanisms involved in the anticancer activity of TTs, but especially of BA, are well known and involve BA. The most important is the BA’s ability to induce apoptosis [30]. These are the extrinsic and intrinsic pathways involved in apoptosis, via the death receptor and the mitochondrial pathway. The extrinsic pathway is activated by the binding of CD95/APO-1/Fas-ligand to CD95R/APO-1R/FasR of the tumor necrosis factor receptor superfamily, while the intrinsic pathway leads to caspase activation, which constitutes a terminal event in apoptosis. While the commonly used anticancer agents either trigger the Fas-associated death receptor pathway of apoptosis or induce cellular stress such as cytokine withdrawal or DNA damage, BA directly induces mitochondrial damage, overcoming the resistance of tumor cell mitochondria [65].

The anticancer therapy using BA is based mainly on the initiation of the mitochondrial pathway of apoptosis [14], while cells and normal tissues are not affected. This offers an important alternative when other chemotherapy based on synthetic drugs fails to circumvent drug resistance in human cancers. Studies on the BA effect are yet to be developed by networks of clinical trial groups supported by the U.S. National Cancer Institute [21].

The antitumor effects of OA and UA on adult T-cell leukemia MT-4 cells were confirmed by dose-dependent inhibition. The UA-treated cells showed caspase 3/7 and caspase 9 activation and mTOR and PDK-1 inhibition following mitochondrial dysfunction [71].

Triterpene-BODIPY adducts belonging to the class of mitocans were recently synthesized and screened for their cytotoxic activity on human non-malignant vs. tumor cell lines (uptake, transport, and metabolization), with the fluorescent 3-O-acetyl-BA-BODIPY conjugate being cytotoxic for human breast adenocarcinoma cells MCF7 but not cytotoxic for all other cell lines [72]. When associated with lipophilic cations (e.g., F16 and TPP), they become more effective as anticancer agents due to their ability to accumulate in the mitochondria of cancer cells and cause ROS bursts and cell death. This also applies to triterpenic acids of the ursane and oleanane types.

Meanwhile, several other BA derivatives, e.g., 7b-hydroxy, 6a,7b-dihydroxy, and 1b,7b-dihydroxy BA, obtained by microbial transformation of BA, have been tested for anticancer activity but showed reduced biological activity [72], as well as oleanane- and ursane-type triterpenes of similar structures, which showed inferior activity relative to BA on a mouse melanoma cell line [73].

The introduction of a heterocycle into the pentacyclic triterpene molecule or the oxidation of the pentacyclic triterpene rings can significantly enhance biological activity. A recent study [74] described the synthesis of 37 derivatives of OA, UA, and glycyrrhetinic acid linked with L-phenylalanine or L-proline. The structural modification, especially for two glycyrrhetinic acid derivatives, induced significant antitumor activities in vitro (against SK-OV-3, MGC-803, T24, HeLa, and A549 tumor cell lines and normal human hepatocytes HL-7702) via apoptosis (decrease of mitochondrial membrane potential, activation of caspase-3/8/9, and inhibition of proteasome activity). In vivo, these derivatives increase solubility and bioavailability due to their link to polar amino acid molecules, which are not only cellular signal molecules but also regulators of gene expression and the protein phosphorylation cascade [75].

Triterpenoid (BA, UA, and OA)-based ionic compounds (which improve their solubility and their permeability through biological membranes) were synthesized and tested for structure–antitumor activity [76]. The ionic derivatives contained these triterpenoids covalently linked to the N-methylimidazole cation, and various nitrogen-containing compounds (pyridine, piperidine, morpholine, pyrrolidine, triethylamine, and dimethylethanolamine) and counterions (BF4-, SbF6-, PF6-, CH3COO-, C6H5SO3-, m-C6H4(OH)COO-, and CH3CH(OH)COO-) have been synthesized. The cytotoxicity was tested on various tumor (Jurkat, K562, U937, HL60, and A2780) and normal (HEK293) cell lines. IC50, signaling, and apoptosis induction on the cell cycle and mitochondria were determined by enzyme immunoassay and flow cytometry. OA-based ionic compounds (ionic derivatives) proved to be the most effective inducers of apoptosis, explained by their higher solubility and better permeability through the biological membranes. Their mitochondrial apoptosis pathway involved the loss of mitochondrial cytochrome c and apoptosis via a p53-dependent mechanism.

Angiogenesis is a key process involved in tumor metastasis and in developing tumor resistance to cytotoxic therapy. There is little data on B as an anti-angiogenic agent; a preliminary study aimed to evaluate the cytotoxic effect on three cancer cell lines: HeLa (cervix adenocarcinoma), MCF7 (breast adenocarcinoma), and A431 (skin epidermoid carcinoma) and their associated apoptotic mechanisms. The angiogenic effect of B was evaluated using morphological and immunohistochemical techniques by double fluorescence staining, indicating that at higher concentrations, the chicken embryo chorioallantoic membrane permeability is enhanced, while at lower concentrations there is evidence for nuclear fragmentation. B induced the reduction of newly formed capillaries, especially in the mesenchyme, by targeting the normal function of endothelial cells. In vitro results proved the superior specificity of B on cervical cancer cells, followed by skin cancer cells [70].

Further studies on the anticancer mechanisms of BA derivatives are expected and will certainly lead to the development of new drugs based on TTs, a clinically approved anticancer therapy.

## 4. Nano-Delivery Vehicles (Liposomes, Nanolipids, and Other Complexes)

In the last few decades, nanotechnology has been extensively studied and exploited for cancer treatment since nanoparticles can play a significant role as a drug delivery system. Compared to conventional drugs, nanoparticle-based drug delivery has specific advantages, such as improved stability and biocompatibility, enhanced permeability and retention, and precise targeting. A recent review reveals the methods of preparation, characterization, and application of several nanoparticle drug delivery systems [77]. Recently, updated information was reported about nanoformulations for the delivery of TTs in anticancer therapies [78]. Multiple forms of lipid-based nanoparticles exist, including liposomes, solid lipid nanoparticles (SLNs), nanostructured lipid carriers (NLCs), microemulsions, nanoemulsions, phytosomes, lipid-coated nanoparticles, and nano-assemblies.

The most known nano-delivery vehicles are liposomes, which are bilayer spheres that incorporate hydrophilic drugs in the aqueous nucleus or lipophilic drugs in the membrane. Liposomes provide enhanced efficacy and reduced toxicity for different anticancer agents and are adequate for enhanced permeability and retention in tumor vessels. Liposomal anthracyclines, such as Doxorubicin (DOXO), proved high efficiency after encapsulation, with significant anticancer activity and reduced cardiotoxicity compared to free DOXO. New versions with prolonged circulation and higher efficiency, such as the PEGylated (Stealth) liposomal DOXO, were produced [79]. A new generation of liposomal delivery systems includes immunoliposomes for improved molecular targeting and tumor recognition, e.g., PEGylated liposomal DOXO (commercially known as Doxil^®^ and Caelyx^®^), which achieve prolonged circulation with a terminal half-life of 55 h in humans.

The development of nano-delivery lipid-based particles is another emerging field of lipid nanotechnology with several potential applications in drug delivery. Due to their unique size-dependent properties, lipid nanoparticles offer the possibility of developing new therapeutics since Solid Lipid Nanoparticles (SLNs), due to their composition of physiologically similar lipids, are well tolerated. Nanostructured lipid carriers (NLCs), including a blend of solid and liquid lipids that results in a partially crystallized lipid system [80], represent a second generation of SLNs [81]. Their ability to incorporate drugs, especially lipophilic ones, offers the possibility of creating new prototypes in drug delivery with increased bioavailability of lipophilic drugs along with controlled and site-specific drug delivery. They have many advantages over SLNs, such as enhanced drug loading capacity, improved stability, drug release modulation flexibility, and the ability to prevent drug leakage. NLCs have numerous applications in both the pharmaceutical and cosmetic industries due to their ease of preparation, the feasibility of scale-up, biocompatibility, non-toxicity, enhanced targeting efficiency, and the possibility of site-specific delivery via various routes of administration.

Different procedures to obtain NLCs were elaborated, but the most widely applied procedures are the melt-emulsification and ultrasonication methods [82], and new data about their structure, preparation, and application were recently reviewed [83,84,85,86,87].

Figure 2 presents the different structures of some representative lipid nanoparticles, e.g., liposomes and NLCs, for the incorporation and delivering of anticancer agents like hydrophilic Doxorubicin (DOXO.HCl) or lipophilic (DOXOn), betulinic acid (AB), and triterpenoid extracts (TT).

As mentioned before, TTs and their semi-synthetic derivatives possess cytotoxic effects on various tumor cell lines. However, due to their low solubility and bioavailability as free molecules, their medicinal applications are rather limited. The use of various nanotechnology-based drug delivery systems is a rapidly developing approach, and the drug delivery systems to be applied for TTs are welcome. Their incorporation into organic nanoparticles (dendrimers, polymer matrices, or complexation with carbohydrate nanoparticles) without covalent bonding, as well as encapsulation and emulsification into nanoparticles, was recently reviewed [88].

In addition, nanoparticle-based drug delivery systems have been shown to play a role in overcoming cancer-related drug resistance. The mechanisms of cancer drug resistance include overexpression of drug efflux transporters, defective apoptotic pathways, and a hypoxic environment. Nanoparticles targeting these mechanisms can lead to an improvement in the reversal of multidrug resistance. Recently, the role of nanoparticles and hybrid nanoparticles in immunotherapy, which plays a more important role in cancer treatment, as well as their function in reversing drug resistance, was also investigated [89].

The solubility of betulinic acid in the microemulsion system of methyl acetate/Tween 80: BRIJ30/H_2_O was reported [20], as was the use of B nanoemulsion for the treatment of mouse skin cancer [90]. Recently, free TTs have successively been found to self-assemble or co-assemble into self-contained nanostructures with enhanced water dispersibility and oral bioavailability, which seems to be an efficient processing method for increased oral efficacy. Of particular interest, formulating them into nanostructures can also be introduced as functional delivery carriers for bioactive compounds or drugs with various advantages, such as improved stability, controlled release, enhanced oral bioavailability, synergistic bioactivity, and targeted delivery [16]. Interestingly, it was shown that some triterpenoids like B and UA form self-assemblies with a 3D structure or a bilayer structure and can incorporate different fluorophores as well as anticancer agents (DOXO) at physiological and lower pH, making them useful as drug delivery vehicles [91,92].

Different in vitro and in vivo systems used to deliver triterpenoid molecules, either free or incorporated into different delivery systems, mainly liposomes and nanolipid complexes, are presented in Table 3.

## 5. New Vehicles for Anticancer Theranostics: Microbubbles and Sonoporation-Assisted Delivery

Theranostics is currently being developed as a combined diagnostic–therapeutic clinical platform, whereas integrated anticancer therapy systems involve diagnostic technologies using ultrasound (US), magnetic resonance, or computed tomography. US molecular imaging represents a step forward in imaging technologies, including treatment and surgery, through molecular biology knowledge and nanotechnology. Nowadays, there is a growing interest in the exploitation of diagnostic US, merging US advantages (e.g., non-invasiveness, short and efficient imaging, low cost) with theranostic applications. The diagnostic potential of US includes angiogenesis, atherosclerotic plaque investigation, and the identification of inflammation markers. The recent development and engineering of US contrast agents (USCA), based on gas-filled microbubbles alone or combined with nanoparticles, provide new insights into early tumor detection via angiogenesis, targeted drug delivery, early responses to molecular therapies, and in situ therapy. This theranostics approach influences novel strategies for diagnosis, drug development, and therapy in oncology [125,126,127,128].

Microbubbles (MB) are gas-filled vesicles with sizes of 1 to 10 µm, stabilized by a lipid layer or protein shell, and enhance the contrast by acting as resonators driven by the compressibility of their gaseous core [127,128,129]. In addition to their role as contrast agents, microbubbles are increasingly acknowledged as theranostic agents in the field of drug delivery [130]. They can stimulate drug uptake in cells and tissues through US-induced cavitation, which involves the expansion and compression of the microbubbles’ gaseous core in response to the ultrasound field, inducing mechanical, thermal, and chemical effects [127,128]. A recent review focused on imaging, delivery, ablation, and theranostics with Focused Ultrasound [FUS] as an alternative to surgical intervention or management of malignant growth. Recently, it was reviewed the use of different materials and contrast agents, colloidal or nanoparticulate agents sensitizing FUS, e.g., microbubbles, phase-shift emulsions, hollow-shelled nanoparticles, or hydrophobic silica surfaces, as well as acoustically active hydrogels and membranes for new applications in acoustics, seeking new agents to improve their efficacy [131]. The nanoparticle-loaded MBs increase the cellular uptake of nanoparticles compared to the co-incubation of nanoparticles and MBs [132]. To mention some advantages of using microbubbles as USCA: drug delivery in proximity to its target, a smaller dose of the drug is required compared to the conventional US method used for antineoplastic drug delivery. By attaching various ligands (liposomes and nanolipids) to microbubbles, targeted drug delivery is significantly improved. Whereas free drugs often possess harmful side effects, their encapsulation in microbubbles and subsequent local release, deposition, and potentiation in the target tissue by ultrasound triggering improve the therapeutic index, lower the incidence of adverse events, and achieve successful therapy. Therefore, the use of microbubbles as a tool for drug delivery enhancement has enormous clinical potential, especially in oncology and vascular applications.

Lipid-shelled C_4_F_10_ microbubbles as such, filled with anticancer drugs and free or decorated with drug-incorporated liposomes via covalent thiol–maleimide or avidin–biotin linkages, were obtained in the last decade. Mainly, DSPE lipid anchors on the end of the conjugate and is embedded in the liposome bilayer and MB monolayer shell. Targeted delivery of gas-filled microspheres and contrast agents for US imaging has been reported since 1999 [133,134]. Liposome–MB conjugates are yet to be produced at a larger scale and are considered better targeted drug delivery vehicles compared to microbubbles alone, with liposomes of 200 nm diameter being attached to the microbubble surface preferably by chemical avidin–biotin interaction [135,136].

Figure 3 shows the general protocol (main experimental steps) for delivering anticancer drugs in vivo, using mice inoculated with tumor cells (1). The nanocomplexes of liposomes or NLC charged with BA, TT or DOXOn are linked chemically by PEG2000 and avidin–biotin to MB, are administered i.v. (2). By sonoporation (3), the MB-linked complexes delivers the nanocomplexes, targeting the tumor icells (4). 

Most studies were developed using DOXO embedded in MBs as DOXO-liposome-MB, which allowed US-triggered killing of melanoma cells in vitro even at very low doses of DOXO [128]. Concerning TT molecules, like BA and B, very few data are available concerning their encapsulation in microbubbles or in conjugates of MB-Liposomes/nanolipids containing TTs. Nevertheless, as mentioned in Table 2 and in recent reviews, new and different nanoformulations were recently reported [6]. Polyethylene glycol (PEG2000), a widely used excipient in microbubble formulations known to improve stability through various mechanisms, induces a decrease in the cell membrane lipid order, increasing membrane fluidity. A balance between lipid/PEG 2000 concentrations is required to ensure stability improvement without compromising the acoustic properties of the formulation [137,138]. PEGylated lipids (liposomes or NLC) act as emulsifiers that prevent phase separation and microbubble coalescence and provide a steric brush to prevent plasma protein adsorption on microbubble surfaces and protect them from opsonization and premature removal. Such a steric brush can also hide ligands that are covalently attached to microbubbles [139,140]. The current applications in drug delivery for systems and shell compositions and preparation strategies that improve the monodispersity of MB or lipid-coated microbubbles (size range of 1–10 µm) while retaining ultrasound responsiveness were recently reviewed [141,142]. The concentration of nanoparticles has a significant impact on the stability of microbubbles, and loading an appropriate amount of nanoparticles is helpful in improving the stability of microbubbles. The results also show that nanoparticle-loaded microbubbles with a size distribution in the range of 120–200 μm can be prepared under optimal conditions [143]. The short circulation time of microbubbles in vivo is mainly due to the fast hepatic clearance of microbubbles by Kupffer cells. Drugs can be either injected next to the MBs or loaded directly onto the microbubbles, resulting in drug-loaded MBs. High acoustic pressure and long pulses are required for the sonoprinting of nanoparticles on cells.

Sonoporation has increasing applications in therapeutic US for cancer therapy [144,145,146], helping MBs to cross biological barriers due to the mechanical effects of cavitation by US-triggered MBs-based drug delivery strategies for in vivo expanding and compressing the MBs’ gaseous core as a response to the US field. The use of drug-loaded microbubbles to stimulate drug release only in the ultrasound-treated region is valid only when the drugs remain firmly attached to the MB shell. In the case of nanoparticle-MB, this largely depends on the stability of the linker molecules that are used to couple nanoparticles to the microbubble shell. However, the efficacy of microbubble-enhanced drug delivery to tumors is also very likely to depend on tumor type. Repeated injections of microbubbles have been shown to be effective and ameliorate therapeutic outcomes, especially when microbubbles can be attached to targeting ligands. They can be used as effective agents for drug delivery but require further clinical trials to fully evaluate their potential and feasibility in humans [147]. The use of MBs and MBs-nanoligand conjugates in combination with sonoporation has been shown to be an effective method to deliver different anticancer agents to the tumor tissue in preclinical models and relevant clinical trials that have been conducted [148].

Although, according to a recent study [149], conventional lipid-based MBs have poor drug encapsulation efficiency and polymer-based MBs show weak capability in contrast imaging and ultrasound-triggered drug release, novel types of multiporous lipid/PLGA hybrids with designed elasticity of the bubble shells were reported. Opens new ways for investigation leading to better performance in US contrast imaging and US-triggered MB-nanoparticle delivery in vitro and in vivo. Until now, DOXO-loaded lipid/PLGA-MBs have achieved high drug encapsulation efficiency in tumor-bearing mice, with enhanced tumor growth inhibition effects compared with free Dox and Dox-lipid/PLGA MBs without ultrasound. This study provides an innovative multifunctional platform for MBs for ultrasound contrast imaging and drug delivery applications. 

Concerning the delivery of TTs by MB–nanolipid conjugates combined with sonoporation in vitro and in vivo, our group is encouraged by preliminary results [150].

## 6. Conclusions

The class of pentacyclic triterpenoids (TTs) showed interesting biological activity, especially anticancer activity, according to the updated studies in vitro or in vivo. on mechanisms and targeted delivery. The increasing interest in their mechanisms of action and application in preclinical and clinical studies has been noted over the last decade. 

Considering the physical and chemical properties of botulin and betulinic acid (the most studied molecules), as well as oleanolic/ursolic acids and derivatives, their biological and pharmacological activity was reviewed. Special attention was given to their antiviral and especially anticancer activity, considering the recent scientific and preclinical investigations. The specific anticancer mechanisms, structure-related activity, and efficiency as free molecules or nano-encapsulated formulas (as liposomes or nanolipid complexes) were discussed. A special approach was given to liposome-TTs and nanolipid-TTs complexes to be linked to microbubbles, known as contrast agents in ultrasonography. This represents a step forward to improve their antitumor efficiency through their inclusion in supramolecular conjugates, which link microbubbles to nanolipid-TTs by chemical affinity (e.g., avidin–biotin or other connection) to be delivered to target cells via sonoporation. The production of such supramolecular conjugates to deliver the drugs to target cells via sonoporation represents a new scientific and applicative direction to improve TT efficiency, considering their limited availability as free, lipophilic molecules. Relevant and recent examples of in vitro and in vivo studies, as well as the challenges for the next steps towards the application of these complex delivery systems to tumor cells, were discussed, as were the challenges for the next steps towards the application of their targeted delivery to tumor cells, opening new directions for innovative nanotechnological solutions.

## Figures and Tables

**Figure 1 ijms-24-12923-f001:**
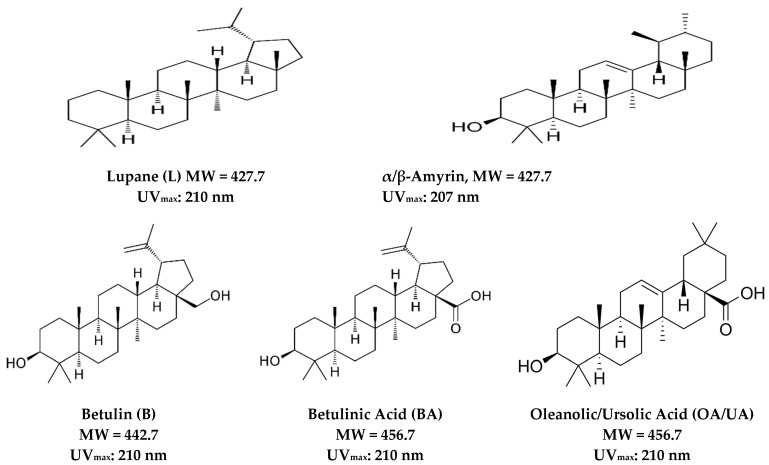
Most representative pentacyclic lupane-type triterpenoids: structure, molecular weight [MW], and maximal UV absorption.

**Figure 2 ijms-24-12923-f002:**
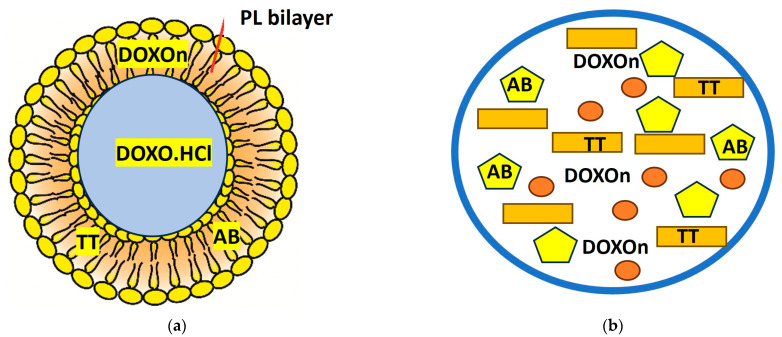
Generic structures of the lipid nanoparticles: liposomes (**a**) and NLCs (**b**) used to incorporate and deliver anticancer agents like Doxorubicin in a hydrophilic (DOXO) or lipophilic (DOXOn) form, betulinic acid (BA), and TTs.

**Figure 3 ijms-24-12923-f003:**
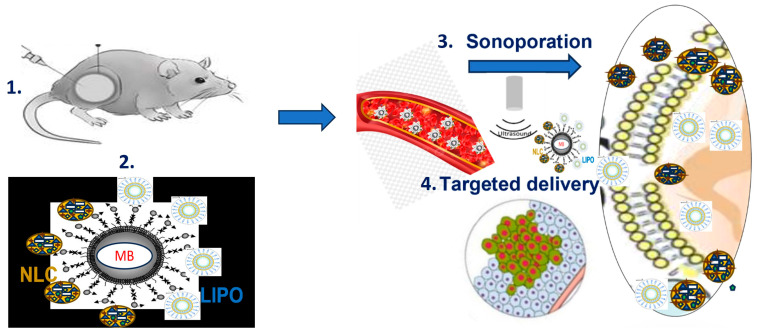
**Experimenta;l steps for the delivery of nanocomplexes (liposomes or NLC charged with BA, TT or DOXOn.** 1: mice inoculated with tumor cells. 2: intravenous administration of liposomes or NLC complexes charged with BA, TT or DOXOn and linked chemically to microbubbles (MB). 3. Sonoporation. 4.The broken MB delivers the nanocomplexes, targeting the tumor icells.

**Table 1 ijms-24-12923-t001:** Effects (targets and mechanisms) of different TTs, evaluated in vitro and in vivo.

Molecules	Effects: Targets and Mechanisms	References
BA	Inhibition of cyclic AMP-dependent protein kinase, sulfonylureas, stromelysin, and collagenase	[3]
B and BA	Differential effects on cytokine production in human blood cell cultures	[26]
OA and derivatives	Anticancer activity, target profiling, and mechanisms of action	[27]
L and derivatives	Inhibitors of nitric oxide and prostaglandin E2	[28]
B, BA, and L	Stimulus-induced superoxide generation and tyrosyl phosphorylation of proteins in human neutrophils	[29]
TTs	Induction of the mitochondrial pathway of apoptosis, especially in melanoma cells	[30]
TTs (*Euonymus alatus twigs)*	Bioassay-guided isolation and antiproliferative effects	[31]
B	Protects HT-22 hippocampal cells against ER stress by induction of heme oxygenase-1 and inhibition of ROS production	[32]
BA	Cytotoxic against tumor cells and reduced cytotoxicity against normal dermal fibroblasts and blood lymphocytes	[33]
B extract (*Acacia auricularis*)	Multi-target inhibitor of protein kinases, a modulator of mitogen-activated protein, affects the kinase pathway (ABL1 inhibitor)	[34]
TTs from *Rhus chinensis Mill*	Inhibition of enzymes in glycolytic enzymes involved in glutaminolysis	[35]
BA	In vitro: significant increase of intracellular-free calcium and a slight decrease in cell viability (low cytotoxicity at relatively high therapeutic doses)	[1,36]
B and BA esters of conjugated linoleic acid	Low cytotoxic activity in vitro and in vivo	[37]
B	Promotes differentiation of human osteoblasts in vitro. Osteoinductive effect on the hFOB Cell Line by Activation of JNK, ERK1/2, and mTOR Kinases	[38]
B, BA, L, OA, and UA	Effects on glucose absorption and uptake, insulin secretion, diabetic vascular dysfunction, retinopathy, and nephropathy	[39]
UA and Dev-UA Euscaphic acid extracts of *Potentilla atr.*	In vitro antiproliferative activity and antiestrogenic activity (high binding affinity towards estrogen receptor-*α* and decreased cell growth) by downregulating the expression of mRNA	[40]

**Table 2 ijms-24-12923-t002:** Recent data related to the anticancer effects of BA and B, as shown by different in vitro and in vivo experiments.

Experiments	Results	Ref.
BA action on melanoma cells in vitro and in vivo against TPA-induced tumors and ovarian and melanoma xenografts in mice	Induction of p53 upregulation in metastatic melanoma cells results in surface blebbing and cytoplasmic shrinking and the formation of characteristic DNA fragments, which are indicative of apoptosis. Without toxic effects at a concentration of 500 mg/kg but significant inhibition of tumor development at 5 mg/kg	[1]
BA action on melanoma cells via the mitochondrial pathway of apoptosis, according to former experimental observations [1] and patent review	Anti-proliferative effect via topoisomerase I and II inhibition in tumor cells (1), mitochondrial apoptosis (2), and anti-angiogenesis (3) in cancer cells	[30]
BA action on melanoma cells in vitro	Induction of changes in the potential of the mitochondrial membrane, activation of caspases, and production of reactive oxygen species	[61,62]
BA distribution across various tissues was administered with a single intraperitoneal dose of 500 mg/kg, as determined by LC–ESI-/MS.	Accumulation of BA (452.2 µg/g) in the tumor and 50% in the liver at a significantly higher concentration but not detectable in the blood	[63]
BA action against different cancer cell types, e.g., acting through induction of apoptosis independent of the cell’s p53 status.	Selective induction of apoptosis in a human neuroblastoma cell line is independent of the cell’s p53 status without affecting normal cells	[33,64]
BA action on cells in vitro	Direct induction of mitochondrial damage, leading to Bax/Bak-independent release of cytochrome C, overcoming the resistance of tumor cell mitochondria	[65]
BA action on liver cancer progression	Inhibition through p53/Caspase-3 signaling activation	[66]
BA’s multifunctional action as an anticancer agent	Effects on JAK/STAT, VEGF, EGF/EGFR, TRAIL/TRAIL-R, and AKT/mTOR	[67]
Water-soluble composites of B dipropionate tested in vitro against ascites carcinoma cells and human lung adenocarcinoma cells	Improvement of antitumor and proapoptotic effects compared with B	[68]
BA effects in vitro	Inhibition of growth factor-induced angiogenesis via the modulation of mitochondrial function also occurs in endothelial cells through an antioxidant mechanism.	[69]
B effects in vitro and in vivo (chicken embryo)	Superior specificity for cervical cancer cells, followed by skin cancer cells. Reduction of newly formed capillaries, especially in the mesenchyme, is possible by targeting the normal function of endothelial cells.	[70]

**Table 3 ijms-24-12923-t003:** Different protocols and results obtained from the delivery of different TTs (as free or incorporated in nanoparticles) as improved delivery systems to test their anticancer effects in vitro or in vivo.

Compounds	In Vitro or In Vivo Systems	Results	Ref.
OA and its derivatives	In vitro: Tumor cell lines MCF-7 and MCF-7/ADR, 1321N1 astrocytoma, Hepatocellular carcinoma, colorectal HCT-116 cells	Inhibition of cell proliferation, tumor cell apoptosis, autophagy, regulating cell cycle proteins, inhibiting vascular endothelial growth, tumor cell migration and invasion, and anti-angiogenesis	[27]
UA, OA, and BA free and modified [exocyclic 2-methylene-3-oxo]	In vitro: K562, A549, and MCF-7 Cell lines Cytotoxicity: MTT	Derivatization intensifies activity. Cytotoxicity and DNA-damaging activity in cancer cell lines in vitro	[52]
BA	Murine melanoma B16 cell line	Apoptotic activity	[93]
B	Fish and murine fibroblasts	High cytotoxicity	[94]
B	In vitro: A431, HeLa, and MCF7 cells In vivo: mouse, CAM assay	Angiogenic inhibitor	[77]
B and Betulonic acid	In vitro: isolated Wistar rat liver mitochondria and liposomes	Activate mitochondrial aggregation Inhibit ADP and DNP synthesis. No effect on membrane permeability.	[95]
Erythrodiol (E)	In vitro: effects on liposomal membrane properties	Interacts with the polar head groups of phospholipids E-Liposomes: spherical and smaller than control	[96]
BA	In vivo: HCC tumor cells, including HepG2, LM3, and MHCC97H In vivo: pulmonary metastasis in mice	Inhibition of HCC growth in vivo Blocked pulmonary metastasis-related proteins, including MMP-2, MMP-9, and TIMP2, without obvious toxicity Increase of the pro-apoptotic protein Bax and cleaved caspase-3; decrease of the anti-apoptotic protein Bcl-2. Reduced the reactive oxygen species (ROS) level	[97]
BA and B	In vitro: canine T-cell and B-cell lymphoma and osteosarcoma cell lines	Anti-proliferative and pro-apoptotic effects (concentration- and time-dependent), stronger for BA compared to B	[98]
B isolated from *Quercus incana*	In vitro: non-small cell lung cancer cells, murine melanoma B16 cell line	Apoptotic and antimetastatic activities	[99]
TTs isolated from *Hibiscus syriacus*	In vitro: breast cancer cells	Apoptosis and inhibition of cell migration	[100]
Lead-BA derivative in a polymeric nanocarrier	In vitro and in vivo colon carcinoma therapeutic efficacy by MTT assay, cell cycle	Apoptosis/antiproliferative activity are significantly increased by the nanocarrier system compared to a free-drug, effective therapeutic agent	[101]
**B** derivatives with ethynyl side chain	In vitro: Liposomes (100 nm) on human cancer cell lines Cytotoxicity: sulforhodamine-B-assay. Apoptosis test with trypan blue dye on A431 and A2780	Encapsulation efficiency of 60% Cytotoxicity Antitumor activity by triggering apoptosis.	[102]
**OA** nanoparticles poly[oligo[ethylene glycol] methyl/methacrylate]-b-poly[oleanolic acid methacrylate]	Encapsulated 10-hydroxycamptothecin to achieve efficient cancer therapy In vitro: breast tumor cells In vivo high antitumor efficiency with low adverse effects on 4T1 mouse breast tumor xenograft	Nanostructures of 100 nm size with good drug stability, loading capacity, and efficiency Drug release up to 132 h Anti-inflammatory and anti-cancer activities. good potential as a platform for drug delivery applications	[103]
**BA** encapsulated in PEGylated Liposomes (LIPO)	Liposomes obtained by ethanol injection are characterized by TEM, AFM, DLS, and FTIR Cell lines: HepG2, HeLa, and U14 In vivo: Female Kunming mice	Liposomes of 142 nm, encapsulation efficiency: 64–95%. Good drug release on PEG-LIPO > LIPO Pegylated-LIPO: better inhibitory effect compared to BA-LIPO or free BA, both in vitro and in vivo Rat liver mitochondrial aggregation; no effect on membrane permeability	[104]
BA-LIPO and LIPO	In vitro: Cancer cell lines A549, sw480 fluorescent marker: rhodamine-phosphatidylethanolamine in the lipid bilayer In vivo: Nude mice xenografted with human colon and lung cancer adm. LIPO/BA-LIPO (5 mg/)mL iv. 3×/week i.v. 50 mg/kg	Liposomes of 1–1.5 microns No toxicity is caused by LIPO Reduced growth of human colon and lung tumors in mice (>50%) Oral administration also led to slow tumor growth	[105]
UA free UA-LIPO	In vitro: Cell culture 9L—rat gliosarcoma; MCF-7 transfected with the luciferase gene In vivo: female nude mice ip. injection, 5 days free UA—23 mg/kg Vs. LIPO	UA-LIPO (0.77 mg/)mL size: 182.7 nm UA-LIPO: Antitumor/antiangiogenic effect in the human breast tumor model but not in the gliosarcoma model	[106]
UA-LIPO	In vitro: breast [MDA-MB-231] and prostate [LNCaP] cancer In vivo: iv mice	UA entrapped: 0.77 mg/mL; stability: 2 months, 4 °C Cytotoxicity: significant inhibition of cancer cell proliferation	[107]
BA-LIPO coated with chitosan [extracted from *Agaricus*) vs. commercial chitosan	FTIR, XRD [X-ray diffraction], DSC [differential scanning calorimeter], DLS [EE: RP-HPLC] Antioxidant activity: DPPH, ABTS	BA-LIPO: smaller size, higher zeta potential. increased antioxidant activity compared to commercial chitosan	[108]
BA-LIPO ± cancer cell membranes	BA-LIPO coated with cell membranes by US compared to control LIPO Characterization: HPLC, TEM, EE, fluorescence microscopy In vitro: HeLa Cell line Viability: MTT, fluorescence	LIPO size: 120 nm (without cell membrane) and 150 nm (with cell membrane) EE: efficiency of 88% on cell toxicity	[109]
BA-LIPO coating of poly-branched Au-Pd bimetallic nanoflowers	NIR photothermal therapy In vitro: HeLa Cell line In vivo model: U14 tumor-bearing mice: intratumoral injection (1.4 mg/kg BA)	BA-LIPO Size: 144.4 nm Photothermal conversion efficiency: 64.6% Tumor inhibition ratio: 91.7%	[110]
BA in nanosized SilCo and PEG-SilCo-coated formulations	In vitro: HepG2 human hepatocellular carcinoma, A549 human lung carcinoma cell Cytotoxicity: Almar blue assay	Size: 20–80 nm SilCo and PEG_SilCo Drug carriers with improved solubility Enhanced antitumor activity using silver, SilCo Significant cytotoxic effect on cancer cell lines	[111]
BA-LIPO	BA 1% in LIPO modified with the biosurfactant mannosylerythritol lipid-A (MEL-A) In vitro: HepG2 cells	Size BA-LIPO: 80 nm The addition of MEL-A significantly promotes cell cytotoxicity and apoptosis, destroying mitochondrial membrane potential	[112]
OA-LIPO modified with octreotide (O)	LIPO is prepared by the ethanol injection method and loaded with synthetic octreotide that mimics somatostatin Size and zeta potential	OA-LIPO: 127 nm OA-LIPO-O: 130 nm ζ potentials OA-LIPO: −20.73 mV; OA-LIPO-O: −1.42 mV OA-LIPO-O: higher inhibition on cell proliferation and cell uptake (for somatostatic receptor-positive A549)	[113]
Ester, 28-O-phosphatidylbetulin (DAPB)	DAPB obtained from hen egg yolk lecithin and B In vitro: lymphocyte subsets In vivo: mice	Immunomodulating effect on lymphocyte subsets and humoral immune response in mice	[114]
Ligustrazine-TT derivatives	In vitro: Cell lines Bel-7402, HepG2, HT-29, Hela, and MCF-7, and Madin-Darby canine kidney	Anti-tumor effect and better antiproliferative actions when coupled with ligustrazine	[115]
TTs extract: UA, OA, BA, L, BA, from *Carlina acanthifolia and C. acaulis*	Individual TT and phenolics identified by LC–MS In vitro: human melanoma cell lines vs. human fibroblasts (control)	IC_50_ higher than 43.2 to 86% micrograms/mL Apoptotic effects	[116]
BA combined with gemcitabine	BA by US-assisted extraction from *Celastrus orbiculatus* ± gemcitabine In vitro: Human pancreatic cancer cell line, non-small-cell lung cancer cell line In vivo: BxPC-3-derived mouse xenograft model.	BA had equivalent cytotoxicity toward gemcitabine-resistant and sensitive cells Synergistic effects of BA and gemcitabine on viability, apoptosis, and DNA double-strand breaks The combination significantly delayed BxPC-3 tumor growth in vivo compared to gemcitabine alone, accompanied by reduced Chk1 expression	[117]
BA, B, and birch bark TTs extract	Physico-chemical comparison In vitro: skin epidermoid carcinoma (A431), breast carcinoma (MCF7), and cervix adenocarcinoma (HeLa) cell lines	Cytotoxic effects on all cell lines	[118]
BA complex with gamma-cyclodextrin	In vitro: mitochondria isolated from the liver of aged rats In vivo tumor development of B164A5 cells	Decrease in in vitro proliferation and in vivo inhibition of mitochondrial respiration Antineoplastic, inhibition of tumor development	[119,120,121]
Cyclodextrin-bonded TTs Extract	In vitro: primary liver cancer cells compared to healthy human hepatocytes	Cytotoxicity at low concentrations is induced by caspase 3/7-mediated apoptosis. Less toxic in healthy hepatocytes	[122]
BA and OA conjugated with per-*O*-methylated-β-cyclodextrin	Human cancer cell lines (MCF-7, BGC-823, and HL-60)	Two OA-derivatives decreased up to 20× the IC_50_ values (6.06–8.47 μM) compared to OA, Induction of the intrinsic apoptosis pathway via the ROS-mediated activation of caspase-3 signaling	[123]
OA, UA, and GA conjugates with cyclodextrins	In vivo (mouse biotherapy)	Induction of cell death through 3 pathways (apoptosis, ferroptosis, and autophagy) Low toxicity	[124]

## Data Availability

Not applicable.

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
