# Peer review of "Pentacyclic Triterpenoid Phytochemicals with Anticancer Activity: Updated Studies on Mechanisms and Targeted Delivery"

_ijms, 2023, doi:10.3390/ijms241612923_

Round 1
Reviewer 1 Report
Pentacyclic Triterpenoid Phytochemicals with Anticancer activity: updated studies on mechanisms and targeted delivery
The review highlights the role of pentacyclic triterpenoids as pharmacological agents such as antibacterial, antiviral, and anticancer. The paper also covered the advanced and presently much focused topic “nano delivery systems” with good literature presentation (Table 2). However, the manuscript needs further modifications to increase its impact among the readers.
Major corrections:
Section 2: Tabulate the physical and chemical characteristics of TTs discussed in the paper.
Section 3. Heading is presented as “Mechanism of action”. However, the entire section appears as an outline or summary. Only one paper outcome was mentioned. Other article references were just mentioned stating their objectivity rather than the outcome of the study. Table 1 presents the effects and mechanisms associated with relevant literature; however, it is better if a few mechanism-based figures were provided as representations.
Section 4 and 5: Tabulate the relevant literature similar to Table 1 focusing on antiviral and anticancer activity of the molecules in discussion.
Section 5: Provide possible anticancer schematic representations.
Table 2 is well documented.
Figure 3: Lacks clarity and representation is poor, must be improved.
Examples specified in most sections only highlight the objective of the reference articles rather than focusing on the outcome of the experiments.
Conclusion should be re-structured.
Minor corrections
1. Line 40 to 42: Remove. Repetitive sentence.
2. Line 52: “but also lupeol [3]”.
3. Line 55: “assuming their action as pharmacological agents and potent anticancer molecules”.
4. Line 60 to 61, 122: “spp.” Should not be italicized.
5. Line 103: “antitumor activity”.
6. Line 119 to 123: Please state the outcome of the study conducted, rather than just stating the purpose of the work.
7. Authors are requested to improve the manuscript by fixing typographical and grammatical mistakes, which are hindering the flow.
Refer the following articles:
https://doi.org/10.1016/j.bioorg.2022.105865
https://pubs.acs.org/doi/abs/10.1021/acs.molpharmaceut.2c00885
https://doi.org/10.1007/s12032-022-01707-x
https://doi.org/10.3390/cancers15030756
https://doi.org/10.1080/10408398.2022.2153238
https://doi.org/10.1016/j.bioorg.2022.106259
https://doi.org/10.1080/14786419.2021.1986495
https://doi.org/10.2174/1389203724666230418123409
Authors are requested to focus on articles published in 2022 and 2023 for improving the manuscript and its impact over the intended audience. The manuscript is well-structured but needs a few schematic representations and tables to further enhance its readability and garner interest among the readers.
Typographical and grammatical errors should be rectified.
Author Response
The answers are attached in a separate file. Thank you agan !!

Reviewer 2 Report
In general, this is a fairly high-quality review related to popular topic. Its distinguishing feature is the description of methods for encapsulating triterpenoids, which can significantly modify their properties and effectiveness as anticancer agents. I have only a couple of comments:
1. Line 231. The authors write about the effects of triterpene-BODIPY adducts. It is important to note here the presence of a whole class of similar compounds called mitocans. These are TTs associated with lipophilic cations (like F16, TPP and others), which are significantly more effective as anticancer agents due to the ability to accumulate in the mitochondria of cancer cells and cause ROS burst and cell death. Moreover, the F16 cation, like BODIPY, can fluoresce, which makes it possible to visualize its accumulation in the cell (for example, in the same MCF-7) and, in particular, in mitochondria. This also applies to triterpenic acids of ursane and oleanane types.
2. Fig. 1. Adjust the image quality.
Author Response
The answers are attached in a separate file. Thank you again !
